# Evaluation of a New Animal Tissue-Free Bleeding Model for Training of Endoscopic Hemostasis

**DOI:** 10.3390/jcm12093230

**Published:** 2023-04-30

**Authors:** Dörte Wichmann, Sarah Grether, Jana Fundel, Ulrich Schweizer, Edris Wedi, Benjamin Walter, Alfred Königsrainer, Benedikt Duckworth-Mothes

**Affiliations:** 1Working Group for Experimental Endoscopy, Development and Training, University Hospital of Tübingen, Waldhörnlestrasse 22, 72072 Tübingen, Germany; sarah.grether@med.uni-tuebingen.de (S.G.); jana.fundel@gmx.de (J.F.); ulrich.schweizer@med.uni-tuebingen.de (U.S.); benedickt.mothes@med.uni-tuebingen.de (B.D.-M.); 2Interdisciplinary Endoscopic Unit, University Hospital of Tübingen, Otfried-Müller-Strasse 10, 72076 Tübingen, Germany; 3Department of General, Visceral and Transplantation Surgery, University Hospital of Tübingen, Hoppe-Seyler-Str. 3, 72076 Tübingen, Germany; alfred.koenigsrainer@med.uni-tuebingen.de; 4General Psychiatry and Psychotherapy with Outpatient Clinic, University Hospital of Tübingen, Calwerstraße 14, 72076 Tübingen, Germany; 5Department of Gynecology and Obstetrics, Diakonissen-Stiftungs-Krankenhaus Speyer, Paul-Egell-Straße 33, 67346 Speyer, Germany; 6Department of Gastroenterology, Gastrointestinal Oncology and Interventional Endoscopy, Sana Klinikum Offenbach GmbH, Starkenburgring 66, 63069 Offenbach, Germany; edriswedi@gmail.com; 7Department of Interventional Endoscopy, Clinic of Internal Medicine I, University Hospital Ulm, Albert-Einstein-Allee 23, 89081 Ulm, Germany; benjamin.walter@uniklinik-ulm.de

**Keywords:** training in endoscopic hemostasis, training endoscopy, hemostasis, variceal banding, clipping for hemostasis

## Abstract

Background: For endoscopists, knowledge of the available hemotherapeutic devices and materials as well as competence in using them is a life-saving expertise in the treatment of patients with acute gastrointestinal bleeding. These competences can be acquired in training on live animals, animal organs, or simulators. We present an animal tissue-free training model of the upper gastrointestinal tract for bleeding therapy. Methods: An artificial, animal tissue-free mucosa and submucosa with the opportunity of injection and clipping therapy were created first. Patches with this artificial mucosa and submucosa were placed into silicone and latex organs with human-like anatomy. Esophageal bleeding situations were imitated as variceal bleeding and bleeding of a reflux esophagitis in latex organs. Finally, a modular training model with human anatomy and replaceable bleeding sources was created. Evaluation of the novel model for gastroscopic training was performed in a multicentric setting with endoscopic beginners and experts. Results: Evaluation was carried out by 38 physicians with different levels of education in endoscopy. Evaluation of the model was made with grades from one (excellent) to six (bad): suitability for endoscopic training was 1.4, relevance of the endoscopic training was 1.6, and grading for haptic and optic impression of the model was 1.7. Conclusions: The creation of a gastroscopic model for the training of hemostatic techniques without animal tissues was possible and multiple endoscopic bleeding skills could be trained in it. Evaluation showed good results for this new training option, which could be used in every endoscopic unit or other places without hygienic doubts.

## 1. Introduction

Gastrointestinal (GI) bleeding is the most frequent cause of an emergency endoscopy, especially for lesions in the upper GI tract with an incidence of emergency endoscopies for bleeding is 36–175 cases per 100,000 residents per year [1,2]. Because of fast diagnosis and location of the bleeding source and the possibility of an effective, endoscopic intervention, emergency endoscopy has become the standard procedure in patients suspected with acute gastrointestinal bleeding [2]. Endoscopists should be well trained and know available hemostatic options in particular materials and devices as well as their applications. Therefore, regular, all-encompassing training of the actual available techniques should be offered to all members, not only to beginners, of an endoscopic unit, especially to mitigate any risk to the patient.

Training of clinical skills plays an important role in medical education [3,4,5]. In many cases, animals are in use for surgical and interventional medical training; however, the use of animals has some weak points. Anatomical differences between humans and animals, challenging hygienic requirements for special rooms and endoscopes, and different or even missing pathologies make training using animals difficult or in the worst case not available. Furthermore, training on animals remains ethically more than questionable. Another problem by using animal tissue for endoscopic training is the possibility of zoonosis. Virtual training or combined virtual training has become available since 2005 [6,7]. However, this training mode does not have a satisfying effect for haptic, and the trainee can only use a restricted number of devices (often only very adapted instruments are available).

Training in emergency patients is difficult not only from an ethical point of view to tolerate and should be avoided if possible. This situation goes along with three moments of stress: stress for the trainee because it is a real emergency, stress for the trainer because he/she must intervene in case of a prolonged endoscopy, and finally stress (and danger) for the patient himself.

Therefore, we consider training on suitable human-like models to be the tool of choice. The following aspects are challenging: the most common causes of bleeding (e.g., gastric and duodenal ulcerations, variceal bleeding, erosive reflux esophagitis) must be identified and imitated by using animal-free materials. The normal application of therapeutic devices must be guaranteed in the model as well as a successful bleeding therapy. Therefore, real therapeutic success should be trainable within the animal-free endoscopic model.

## 2. Materials and Methods

The development and manufacturing of a human-like model for the training of hemostatic skills in flexible endoscopy was the topic of two doctoral theses at the University Hospital of Tübingen. Models for training of the flexible endoscopy known as “Tübingen models” were used as a basis [8], upon which two organ models with exchangeable bleeding pathologies have been built. For these models, 3D data of the stomach and esophagus of patients were used, which were printed as a mold. This so-called negative mold was dipped several times in silicone or latex to create the corresponding organ.

To simulate bleeding, a mechanical piston pump (self-built by Prof. Grund and made available for the project) filled with a red-colored fluid (vegan milk and red food coloring) is used. Vegan milk was chosen because it is also detectable with ultrasound imaging due to its fat and protein content.

(a)Gastric and duodenal ulcers bleeding

An artificial mucosa with a real-life haptic and optic effect is created. To create such an artificial mucosa, the following materials were used: cassava (Praise Export Services Limited, Taifa-Burkina, Accra, Ghana) and tapioca flour (Farmer Brand, Bangkok, Thailand), sodium chloride (VWR International GmbH, Bruchsal, Germany), red-colored water (Brauns-Heitmann GmbH and Co. KG, Warburg, Germany), infusion fluid with 6% hydroxyethyl starch (HES 130/0,4) (Fresenius Kabi Deutschland GmbH, Bad Homburg, Germany), and cotton fibers (dm-drogerie markt GmbH + Co. KG, Karlsruhe, Germany). After thorough mixing at room temperature, the mixture is cooked in a steamer for 15 min. For better release from the steam tray, it is lubricated in advance with Vaseline (dm-drogerie markt GmbH + Co. KG) and the mass is covered with cling film (Melitta Unternehmensgruppe Bentz KG, Minden, Germany). The volume of the mixtures was calculated to result in the characteristic thickness of 3–4 mm of this patch. The resulting patch could be defined by its soft consistency, limited elasticity with tearability, can be cut with high-frequency current, and can be coagulated with Argon Plasma Coagulation (APC) using standard settings (VIO^®^300 D; Art.-Nr. 10140-100, ERBE Elektromedizin GmbH, Tübingen, Germany). The final artificial mucosa is shown in Figure 1.

The ability to swell after injection and application of saline solution or fibrin clue was a challenge to replicate in the artificial submucosa. To simulate such a swelling, in addition to retaining the injected solutions, a submucosal patch was developed by sewing a piece of stocking (40 den) onto a 4.5 × 4.5 cm (gastric), resp. 3 × 3 cm (duodenum) Aquacel Extra cut out (ConvaTec Group plc, Deeside, United Kingdom), leaving a pouch which was filled with a small amount of a superabsorbent (ElaDe.de, Lauffen am Neckar, Germany). Finally, a handmade patch with fixed swelling granulates and sewn stocking was developed. All components of the mucosal and submucosal layers are listed in Table 1.

In the center of the artificial mucosal patch, a bleeding ulcer was imitated using a perforated, small (inner diameter 2 mm, outer diameter 2.4 mm) silicone tube (RCT Reichelt Chemietechnik GmbH + Co, Heidelberg, Germany), connected to the mechanical pump. The mucosal and submucosal patches are inserted to a 3D-printed round sealing cap (Figure 2) with a layer of double-sided adhesive tape in between as separating layer and moisture sealant. Then, the complete patch, so called changeable bleeding-patch, was placed in a manually manufactured silicone gastric or duodenal organ with integrated fittings for the 3D printed artificial bleeding patch holder. For silicon molding, colored Dragon Skin™ 20 with a shore hardness of 20 was used (Smooth-On, Inc. (Macungie, PA, USA)). For easy cleaning, both organs were equipped with zippers, and the duodenal organ was equipped with a sealing cap in the position of the Treitz ligament to prevent leakage during training. Figure 3 shows the endoluminal view on an active gastric bleeding source and possible treatment results with clips.

(b) Esophageal variceal bleeding

An artificial esophagus was made by dipping a gypsum model into fluid latex (Wolff Kunststoffe GmbH, Mörlenbach, Germany). Varices were created using Fimo^®^ soft modelling clay (Staedtler Mars Deutschland GmbH and Co. KG, Nürnberg, Germany) which was then dipped for 2 s into dark blue colored fluid latex (Wolff Kunststoffe GmbH). To simulate bleeding, a small perforation was inserted in the endoluminal position of the varices, connected to the mechanical piston pump. An endoscopic picture of the manufactured variceal bleeding (Figure 4a) and after successful banding (Figure 4b) is shown in Figure 4.

(c) Bleeding out of erosive reflux esophagitis

The latex esophagus is turned inside out, and a red foam material is fixed using glue in the distal esophagus. Then, the esophagus is turned back. To simulate bleeding, a small cannula under the foam coated esophagus was connected to the mechanical pump. In Figure 5, an endoscopic view on the erosive reflux esophagitis is shown.

(d) Modular exchangeable model

The described organs can be attached to the position of the esophagogastric junction. Then, the artificial esophagus can be attached to the position of the hypopharynx. All connections are made with ring connectors for a tight seal. After simulating bleeding, all organs can be removed for manual cleaning. The organs are placed in a plastic torso (ULMER Kunststoffteile GmbH and Co. KG, Sonnenbühl Willmandingen, Germany) filled with foam and connected to a head (CLA—Coburger Lehrmittelanstalt, Coburg, Germany) with a possibility of oral intubation. The artificial organs are placed in a chassis padded with foam. For training, a water-impermeable layer protects the foam from contamination.

(e) Evaluation

A multicentric evaluation in clinical settings was proven and allowed by the Ethics Committee of the University Hospital of Tübingen (515/2019BO2; date of approval: 22 June 2020). Tests were performed in three endoscopic units of different hospitals (i.e., University Hospital of Tübingen, University Hospital of Ulm, and Sana Hospital Offenbach). In sum, 38 physicians of all levels of training tested and evaluated the novel bleeding training model. Beginners were defined as physicians with any or small experience in gastroscopy (*n* = 0–50), without addressing the experience of hemostatic interventions. Different gastroscopes and processors were used. The following therapeutic modalities and devices were used: injection therapy with saline, clipping using Through-the-Scope- (Lockado™Clip, Micro Tech Europe GmbH, Düsseldorf, Germany) and Over-the-Scope-Clips (OTSC-System, Ovesco Endoscopy AG, Tübingen, Germany), variceal banding set (Multiband-Ligatursystem, Micro Tech Europe, Düsseldorf, Germany), and hemostatic powders (EndoClot, EndoClot Plus Inc, Santa Clara, CA, USA). The evaluation was performed using an anonymous questionnaire with school grades, as follows: 1, very good; 2, good; 3, acceptable; 4, sufficient; 5, insufficient; and 6, poor. Figure 6 shows a flowchart of the evaluation procedure.

Two evaluations took place. One of the trainees evaluated the model for the life-like character of the bleeding, and the haptic and one evaluation was performed by the trainers, who ranked the correctness of detected bleeding sources, the correctness of the chosen hemostatic intervention, and the final success of the procedure.

(f) Statistics

The descriptive statistics were conducted with Excel 2019 for Windows 10 (MS Office; Microsoft Corporation, Redmond, Washington, DC, USA).

## 3. Results

A human-like, complete animal tissue-free model for training emergency bleeding gastroscopies was successfully realized. The simulated bleeding sources of upper gastrointestinal bleeding situations and their endoscopic treatment possibilities are listed in Table 2.

The artificial organs with connected drains and bleeding sources are easy to open and to remove for cleaning and exchanging different artificial pathologies. The final training model as used for the evaluation is presented in Figure 7.

The artificial organs are placed on a chassis padded with foam. For training, a water-impermeable layer protects the foam from contamination. The clinical evaluation of the model took place in the endoscopic units of the participating hospitals. The evaluation of the training model was performed between May 2020 and June 2020. In total, 38 participants evaluated the bleeding model in three sessions. The sociometric information on the participants is listed in Table 3; N = 13 participants (34%) were beginners, *n* = 9 participants had an experience of 51–250 gastroscopies, and *n* = 15 participants were experts in the field with more than 250 gastroscopies. Moreover, *n* = 18 participants specified an experience of 0–20 hemostatic interventions.

Participants were not informed about the type of bleeding present (i.e., variceal, ulcerative, or erosive esophageal bleeding) at the beginning of the study. The trial manager asked about the location and type of bleeding while performing the endoscopy. Thirty-two participants replied correctly (84%), whereas four participants replied incorrectly (11%), and two participants did not answer (5%). Then, the participants were instructed to choose the correct therapy option themselves and to perform hemostasis. The correctness of the chosen therapeutic intervention was given in 97%. The success of the therapeutic intervention varied between 100% (hemostatic powder) and 20% (variceal banding).

An evaluation of the trainees for the model was also performed. Table 4 presents the mean school grades for general and specific items of the model sorted according to the educational level of the participants.

## 4. Discussion

An animal tissue-free training model for endoscopic hemostasis of some bleeding events in the upper GI tract was created and evaluated. Currently, training using simulators in medicine is gaining increasing importance for students, nurses, and doctors [9,10,11]. Particularly in the field of emergency medicine, simulator training has long been established. Effective training outside of the daily clinical routine offers the advantage that interventions can be repeated several times without time pressure and stress. According to Haycock et al., practical training, accompanied by a teaching curriculum, is clearly superior to purely theoretical knowledge [12]. The benefits of simulator-based training in medicine, particularly in flexible endoscopy, have already been demonstrated [3,4,5].

Hochberger et al. compared simulator-based training with clinical training alone and showed improved results in the group with simulator-based training in all procedures analyzed [13]. The relevance of adequate training in endoscopy was underlined by the fact that the outcome of a clinical intervention strongly depends on the experience of the examiner [3].

Training in interventional endoscopy using biomodels has been the subject of numerous recent studies. The organs and organ packages integrated into the simulators are mostly from pigs [13,14,15]. The disadvantages of biomodels are as follows: anatomy, which differs from that of humans; high costs of the various procedures with the corresponding logistics because of the animal material; and the risk of zoonosis. Advantages of the biomodels using organ packages are the realistic behavior of the tissue in comparison to patients and the multiple use of clips in different positions of the organs.

The development of a complete animal tissue-free endoscopic training model was challenging. With rechargeable organ modules and numerous bleeding pathologies, a complete training model for the upper gastrointestinal tract (from the mouth to the duodenum) was created. The possible locations of the bleeding are limited by the location of the exchangeable bleeding sources. The development of animal tissue-free mucosal and submucosal layers is essential for imitating gastric and duodenal pathologies. A positive feature of the bleeding model is the realistic anatomy and pathology localization. Should the model be produced industrially, a relevant cost saving could result. Currently, artificial mucosa is manufactured by hand. It can be kept frozen for more than 3 months, but after thawing, the surface structure is slightly altered. Here, a simpler artificial mucosa would be desirable. The presented model complies with all ethical aspects of medical education. It is not practiced on humans, not practiced on pigs, and not practiced on live animals. Furthermore, a realistic atmosphere perhaps in hospitals can be designed for the trainee with human-analog anatomy and different bleeding scenarios.

The composition of the analyzed participants from beginners and experts was chosen to be able to differentiate conclusions about the suitability of the model for different training purposes.

The evaluation of the bleeding model by trainees was positive. In total, 88% of the evaluations corresponded to grades 1 and 2. The overall score was 1.7, and the evaluation of the suitability of the model for training showed an average grade of 1.4; it can thus be concluded that most participants considered the introduced bleeding model as a suitable training model. The visual aspects of the model were rated positively, with a mean grade of 1.5. The evaluation by more experienced endoscopists (specialists/superiors) was slightly less good with a mean grade of 1.8. The haptic effect of the bleeding model was given an average grade of 2.0. The assessment by specialists and senior physicians was more critical than that by residents.

The difference in grading among the participants can be explained by the fact that more experienced doctors have more opportunities for comparison because of their previous knowledge in the clinic and possibly from other training simulators. The correlation between the different educational level of the trainees and success in hemostasis could also provide new insights of reality and further conclusions on the suitability of the model. The realistic haptics and visual appearance of the bleeding model can be well depicted based on the grading.

The introduced training model at the moment has certain limitations; namely the number of interventions is limited. In contrast, in the commonly used EASIE-Trainer with porcine material, more interventions in different positions are possible. However, in the course of further development, additional locations and causes of bleeding can be incorporated, because the model is extremely flexible and allows practically any adaptation at short notice. The second key benefit of the introduced model, in addition to the human-analog anatomy, is the possibility of using common endoscopes in any space (i.e., hospitals or hotels).

## 5. Conclusions

To conclude, the development of a complete animal tissue-free human-like model was successful. The results of the introduced bleeding model provided an adequate training purpose, particularly for beginners in endoscopy.

## Figures and Tables

**Figure 1 jcm-12-03230-f001:**
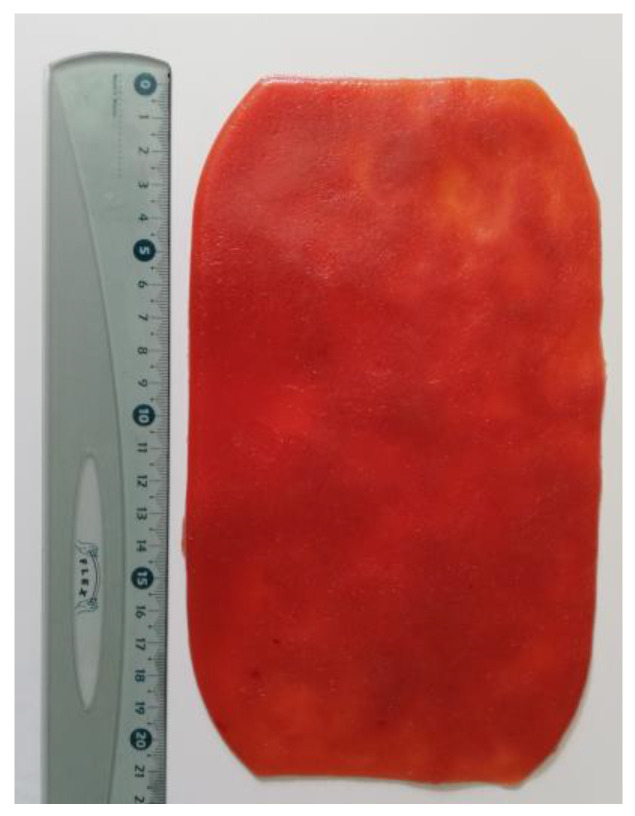
The artificial mucosa patch.

**Figure 2 jcm-12-03230-f002:**
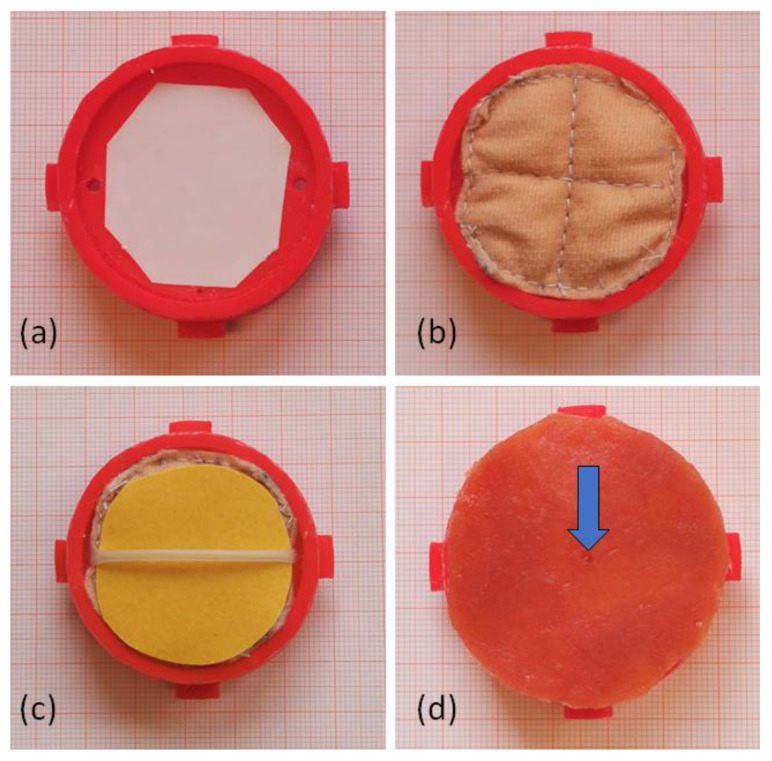
Structure of the bleeding patch (**a**) 3D-printed sealing cap, (**b**) submucosal layer placed into the sealing cap, (**c**) artificial vessel over the submucosa, an impermeable layer separates the vessel from the submucosal layer, (**d**) final changeable bleeding patch with mucosal layer as the top layer, the blue arrow marks the location of the bleeding.

**Figure 3 jcm-12-03230-f003:**
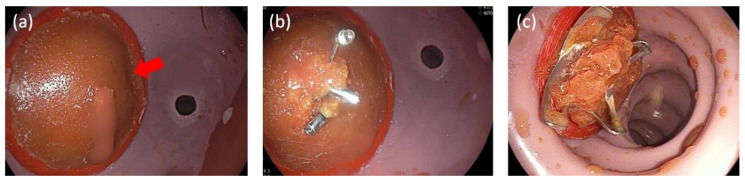
Endoluminal findings by simulating bleedings (**a**) active bleeding in gastric position, the red arrow marks the bleeding; (**b**) successful hemostasis by Through-The-Scope-Clips; (**c**) bleeding patch in duodenal position with successful treatment with an Over-The-Scope-Clip.

**Figure 4 jcm-12-03230-f004:**
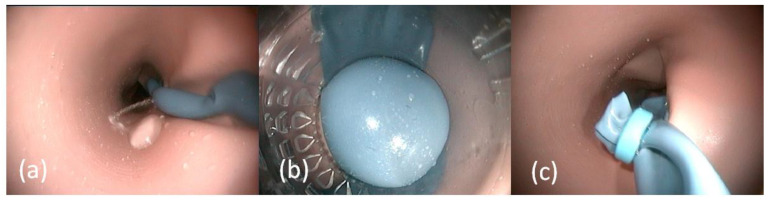
Endoscopic view on the artificial bleeding varices ((**a**), with uncolored water), during banding (**b**) and after successful variceal banding therapy (**c**).

**Figure 5 jcm-12-03230-f005:**
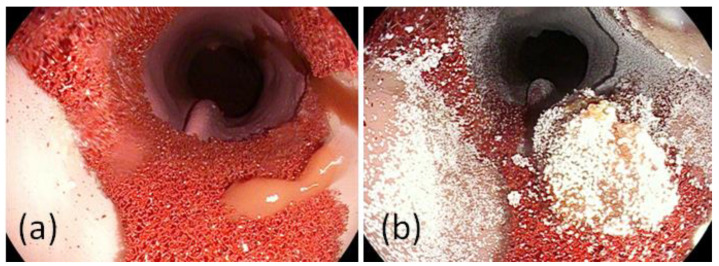
Endoscopic view (**a**) on a bleeding reflux esophagitis and (**b**) after application of a hemostatic powder.

**Figure 6 jcm-12-03230-f006:**
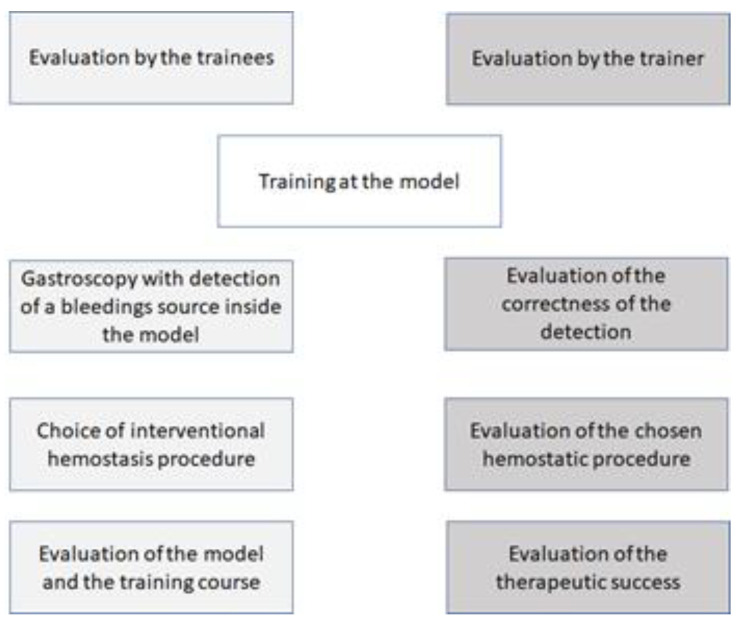
Flowchart of the evaluation procedure.

**Figure 7 jcm-12-03230-f007:**
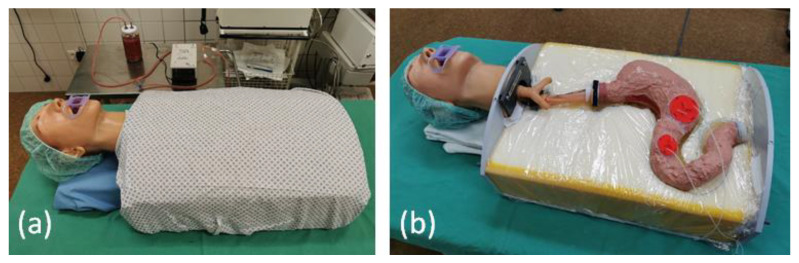
Training model (**a**) final construction with covered model, in the background the mechanical pump with the artificial blood is visible, (**b**) uncovered model, visible are the connections for the bleeding patches.

**Table 1 jcm-12-03230-t001:** List of used materials and manufacturer for the artificial mucosa and submucosa.

Layer	Material	Manufacturer
**Mucosal**	Volulyte 6% infusion fluid	Fresenius Kabi Deutschland GmbH, Bad Homburg, Germany
	Kokonte-Lafu (Cassava Flour)	Praise Export Services Limited, Taifa-Burkina, Accra, Ghana
	Tapioca starch	Farmer Brand, Bangkok, Thailand
	Natriumchloride	VWR International GmbH, Bruchsal, Germany
	Cotton	dm-drogerie markt GmbH + Co. KG, Karlsruhe, Germany
	Red food coloring	Brauns-Heitmann GmbH + Co. KG, Wahrburg, Germany
	Vaseline	dm-drogerie markt GmbH + Co. KG, Karlsruhe, Germany
	Cling film	Melitta Unternehmensgruppe Bentz KG, Minden, Germany
**Submucosal**	Aquacel Extra (10 cm × 10 cm)	ConvaTec Group plc, Deeside, United, Kingdom
	Tights 40 den	dm-drogerie markt GmbH + Co. KG, Karlsruhe, Germany
	Schauch Superabsorber (Art.-Nr. 397)	ElaDe.de, Lauffen am Neckar, Germany

**Table 2 jcm-12-03230-t002:** List of imitated bleeding sources and the offered therapy modalities.

Bleeding Source	Treatment Modality
Variceal bleeding	Ligation therapy
Gastric ulcer bleeding	TTSC, OTSC, injection therapy
Duodenal ulcer bleeding	TTSC, OTSC, injection therapy
Erosive esophagitis with bleeding	Topic hemostatic powder

Abbreviations: TTSC = Through-The-Scope-Clip; OTSC = Over-The-Scope-Clip.

**Table 3 jcm-12-03230-t003:** Sociometric information and characteristics of the participants.

Sociometric Items	Number of Participants (%)
Age	
<30 years	9 (24%)
30–45 years	23 (60%)
>46 years	6 (16%)
Male Sex	26 (68%)
Level of education	
Assistant physician	17 (45%)
Medical specialist	5 (13%)
Senior physician	10 (26%)
no information	6 (16%)
Level of knowledge	
number of elective gastroscopies 0–50	13 (34%)
number of elective gastroscopies 51–250	9 (24%)
number of elective gastroscopies > 251	15 (39%)
no information	1 (3%)
bleeding gastroscopies 0–20	18 (47%)
bleeding gastroscopies 21–50	5 (13%)
bleeding gastroscopies 51–100	14 (37%)
no information	1 (3%)

**Table 4 jcm-12-03230-t004:** Mean grades for general and specific items of the model and endoscopic training.

Parameter	Mean School Grade		
	Total	Assistant Physician (*n* = 17)	Medical Specialists, Senior Physicians and Others (*n* = 21)
Importance of training in endoscopy for bleeding situations	1.6	1.3	1.9
Suitability of the model for endoscopic training for bleeding treatment	1.4	1.2	1.8
Model in general			
Overall grade	1.7	1.4	2.2
Optic	1.5	1.3	1.8
Haptic	2.0	1.6	2.3
Gastric and duodenal ulcer			
Submucosal wheal	1.8	1.7	2.0
Application ground for the TTSC	1.8	1.5	2.4
Application ground for the OTSC	1.4	1.3	1.7
Success of therapy	1.5	1.2	1.7
Erosive esophagitis			
Hemostatic powder	1.6	1.2	2.7
Success of therapy	1.6	1.4	2.7
Variceal bleeding			
Banding ligation	2.3	1.5	2.8
Success of therapy	2.3	1.0	2.8

Abbreviations: TTSC = Through-The-Scope-Clip, OTSC = Over-The-Scope-Clip.

## Data Availability

Data are available from the first author upon request.

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
