# Peer review of "Evaluation of a New Animal Tissue-Free Bleeding Model for Training of Endoscopic Hemostasis"

_jcm, 2023, doi:10.3390/jcm12093230_

Round 1

Reviewer 1 Report

It is an interesting study of the new animal-tissue-free training model.

I have some comments for this issue.

1. I wonder about the reality like as a human and the cost-effectiveness compared to the animal model of this training simulator. Please describe it in the discussion section.

2. Is erosive reflux esophagitis a disease with a high frequency of UGI bleeding? I don't think so. Why did the authors choose this model along with GU/DU and esophageal variceal bleeding?

3. about figure 4, readers can't understand easily what this picture means. If authors want to use this picture, additional pictures (like fig.3b and 3c) are needed.

4. In the results section, it would be better to present the procedure pictures of GU/DU bleeding control. (like fig. 3)

5. In Table 2, there is no explanation of abbreviations.

6. It needs a footnote explaining the meaning of the grade in Table 4.

Author Response

Thank you for the well-meaning review and helpful comments on the manuscript. The following is the point-by-point analysis:

  1. I wonder about the reality like as a human and the cost-effectiveness compared to the animal model of this training simulator. Please describe it in the discussion section.

Thanks for this remark. The following sentences are inserted into the discussion part: A positive feature of the bleeding model is the realistic anatomy and pathology localization. Should the model be produced industrially, a relevant cost saving could result. Currently, the artificial mucosa is manufactured by hand. It can be kept frozen for more than 3 months, but after thawing the surface structure is slightly altered. Here, a simpler artificial mucosa would be desirable.

  1. Is erosive reflux esophagitis a disease with a high frequency of UGI bleeding? I don't think so. Why did the authors choose this model along with GU/DU and esophageal variceal bleeding?

Thanks for this point. The reflux esophagitis could be a bleeding pathology in the upper GI. Of course, the relevance of reflux esophagitis is much less compared to variceal bleeding, but it is a good practice indication for training the application of adhesive substances. See now the new Figure 5.

  1. about figure 4, readers can't understand easily what this picture means. If authors want to use this picture, additional pictures (like fig.3b and 3c) are needed.

Thanks for this comment, we inserted the new Figure 5 with a bleeding reflexive esophagitis and after treatment with hemostatic powder.

  1. In the results section, it would be better to present the procedure pictures of GU/DU bleeding control. (like fig. 3)

Thanks for this remark. We inserted the new Figure 2 with endoscopic findings of a changeable bleeding patch and results after therapy in gastric and duodenal position.

  1. In Table 2, there is no explanation of abbreviations.

Abbreviations are inserted now.

  1. It needs a footnote explaining the meaning of the grade in Table 4.

Thanks for this remark. Footnote is inserted now.

Reviewer 2 Report

The concept is interesting but it appears to be a work in progress

The limitations section needs to be expanded as the numbers are limited and it is apparently proof of concept 

In the discussion section advantages of the animal free model needs to be highlighted

Author Response

The concept is interesting but it appears to be a work in progress. The limitations section needs to be expanded as the numbers are limited and it is apparently proof of concept. In the discussion section advantages of the animal free model needs to be highlighted.

Thanks for these advices. We changed discussion part in a substantial manner.

Reviewer 3 Report

We appreciate the opportunity for peer review. In this study, the authors created a model for learning endoscopic hemorrhage management and investigated the usefulness of training with it for endoscopic beginners and experts. The results seemed to indicate that this training model would be a useful tool for beginners. In addition, we believe that the development of a model that does not require the use of animal material would be very useful. However, in order to accept the paper, we would like to make the following comments.

Major

Methods

1. In the method of evaluation, I think it would be better to describe what and how to evaluate in the method section as a sentence, not just as a figure. In Figure 5, it seems that the items to be evaluated by trainees and trainers are different, however, this is not described in the method.

2. The definition of beginners and experts in the abstract remains unclear in the method. Please clarify this point.

Results

3. There seems to be a discrepancy between the flow of the study described in the flowchart and the explanations given in the results. Please describe the results according to the flowchart or consider a way to describe the results so that readers can easily understand them.

4. Lines 171 - 174 on page 6 should be in the Methods, not the Results.

5. I think it would be easier to understand the results if Table 4 showed the overview of participants, but separated the results for beginners and experts.

6. I think it would be easier to understand the results if the total number listed in Table 4 is the average of 38 participants. Please include the number of assistant physicians and other evaluators. Abbreviations are mentioned in the text, but please include them in the footnote.

Discussion

7. I believe that a brief description of what has been found in this study in the first paragraph will lead to a valuable discussion that will follow.

Minor.

1. In Figure 2, the bleeding should be shown in the figure so that the reader can better understand the usefulness of the model.

2. In Figure 4, it is difficult to see the bleeding point; can you show it with an arrow?

Author Response

We appreciate the opportunity for peer review. In this study, the authors created a model for learning endoscopic hemorrhage management and investigated the usefulness of training with it for endoscopic beginners and experts. The results seemed to indicate that this training model would be a useful tool for beginners. In addition, we believe that the development of a model that does not require the use of animal material would be very useful. However, in order to accept the paper, we would like to make the following comments.

Major

Methods

  1. In the method of evaluation, I think it would be better to describe what and how to evaluate in the method section as a sentence, not just as a figure. In Figure 5, it seems that the items to be evaluated by trainees and trainers are different, however, this is not described in the method.

Thanks for this advice. We inserted some sentences in the method section: Two evaluations took place. One of the trainees, who evaluated the model, the life-like character of the bleeding and the haptic and one evaluation was performed by the trainers, who rank the correctness of detected bleeding sources, the correctness of the chosen hemostatic intervention and the final success of the procedure.

  1. The definition of beginners and experts in the abstract remains unclear in the method. Please clarify this point.

Thanks for this comment. Beginners are defined as physicians with any or small experience in gastroscopy not only for bleeding situations. N=13 participants (34%) were beginners defined with an experience of n= 0-50 gastroscopies. N=9 participants had an experience of 51-250 gastroscopies and n=15 participants were experts in the field with more than 250 gastroscopies. Moreover this, n=18 participants specified an experience of 0-20 hemostatic interventions. We inserted this sentence into the methods section: Beginners were defined as physicians with any or small experience in gastroscopies (n=0-50), without addressing the experience of hemostatic interventions.

Results

  1. There seems to be a discrepancy between the flow of the study described in the flowchart and the explanations given in the results. Please describe the results according to the flowchart or consider a way to describe the results so that readers can easily understand them.

Thanks for this advice. We changed the result section: Participants were not informed about the type of bleeding present (i.e., variceal, ulcerative, or erosive esophageal bleeding) at the beginning of the study. The trial manager asked about the location and type of bleeding while performing endoscopy. Thirty-two participants have replied correctly (84%), whereas four participants replied wrong (11%), and two participants did not answer (5%). Then, the participants were instructed to choose the correct therapy option themselves and to perform hemostasis. The correctness of the chosen therapeutic intervention was given in 97%. The success of the therapeutic intervention varied between 100 % (hemostatic powder) and 20% (variceal banding).

An evaluation of the trainees for the model was performed also. Table 4 presents the mean school grades for general and specific items of the model sorted according to the educational level of the participants.

  1. Lines 171 - 174 on page 6 should be in the Methods, not the Results.

Thanks for this advice, the paragraph is now in the methods part.

  1. I think it would be easier to understand the results if Table 4 showed the overview of participants, but separated the results for beginners and experts.

Yes, that is the content of Table 4.

  1. I think it would be easier to understand the results if the total number listed in Table 4 is the average of 38 participants. Please include the number of assistant physicians and other evaluators. Abbreviations are mentioned in the text, but please include them in the footnote.

Thanks for your advice. Yes, in Table 4 the average of all participants (n=38) are listed. The total number is included now. A footnote is inserted also.

Discussion

  1. I believe that a brief description of what has been found in this study in the first paragraph will lead to a valuable discussion that will follow.

Thanks for this advice. We inserted the sentence: “An animal tissue free training model for endoscopic hemostasis of some bleeding events in the upper GI tract was created and evaluated.”

Minor.

  1. In Figure 2, the bleeding should be shown in the figure so that the reader can better understand the usefulness of the model.

Thanks for this comment. We changed Figure 2 according to your suggestions. Moreover, we inserted a new Figure (3) with endoscopic findings of a bleeding changeable patch and the patch after treatment in gastric and duodenal position.

  1. In Figure 4, it is difficult to see the bleeding point; can you show it with an arrow?

Thanks for this point. We changed the new Figure 5.

Round 2

Reviewer 3 Report

The presented manuscript is revised adequately.